# Selective transport of water molecules through interlayer spaces in graphite

Lalita Saini [1], Siva Sankar Nemala [1], Aparna Rathi [1], Suvigya Kaushik [1] & Gopinadhan Kalon [1,2 ✉]

Interlayer space in graphite is impermeable to ions and molecules, including protons. Its controlled expansion would find several applications in desalination, gas purification, high-density batteries, etc. In the past, metal intercalation has been used to modify graphitic interlayer spaces; however, resultant intercalation compounds are unstable in water. Here, we successfully expanded graphite interlayer spaces by intercalating aqueous KCl ions electrochemically. Our spectroscopy studies provide clear evidence for cation-π interactions explaining the stability of the devices, though weak anion-π interactions were also detectable. The water conductivity shows several orders of enhancement when compared to unin-tercalated graphite. Water evaporation experiments further confirm the high permeation rate. There is weak ion permeation through interlayer spaces, up to the highest chloride concentration of 1 M, an indication of sterically limited transport. In these very few transported ions, we observe hydration energy-dependent selectivity between salt ions. These strongly suggest a soft ball model of steric exclusion, which is rarely reported. These findings improve our understanding of molecular and ionic transport at the atomic scale.

[1] Discipline of Physics, Indian Institute of Technology Gandhinagar, Gandhinagar, Gujarat, India. [2] Discipline of Materials Engineering, Indian Institute of Technology Gandhinagar, Gandhinagar, Gujarat, India. ✉email: gopinadhan.kalon@iitgn.ac.in

Selective transport of molecules and ions is ubiquitous in nature[1,2]. For example, aquaporins transport water molecules selectively while blocking all the ions, including protons[3]. Mimicking biological channels would result in highly efficient filtration systems. Several low-dimensional materials, including nanopores, nanotubes, channels, and laminates[1,4–8] have been investigated for this purpose. Of these, two-dimensional materials were found to be very attractive due to their thinness, ability to form heterostructures, and the possibility to tune their interlayer spaces[9–12]. However, several challenges limit their potential usage, these include uncontrollable functionalization[13], swelling in aqueous solutions[14], and multiple processes involved in the device fabrication. Moreover, the membrane architecture formed from the assembly of exfoliated individual layers is highly susceptible to large-scale defects and pinholes.

To address this problem, we chose high-quality single-crystalline graphite samples. It is reported that, at thermal energies, no ions can permeate through graphite, including small-sized protons[15]. The weak van der Waals interaction of the layers in graphite offers a possibility to modify its interlayer space. The controlled expansion of the layers would allow selective transport of molecules and ions. The intercalation of metal atoms such as lithium, calcium, etc. in graphite results in intercalation compounds[16], which have been well studied. However, these compounds are exothermic in nature and decompose while in contact with water. This prevents their utility in water-related applications.

In our approach, we utilized a geometry and intercalant very different from the conventional intercalation. A thin crystal of graphite is placed in between two reservoirs of aqueous KCl solutions in such a way that the graphite has space to expand upon ion intercalation (Fig. 1a). There are several reasons to choose KCl ions for the intercalation; the hydrated radius of $K^+$ and $Cl^-$ is the smallest among all the salt ions[17], and the intercalation is expected to provide an interlayer distance sufficient to let water molecules pass through while rejecting salt ions. The ionic nature of the solution helps the intercalation to be done electrochemically, which is fast, efficient, and a room-temperature process. It is worth mentioning that several salt ions were utilized recently to control the interlayer distance of graphene oxide membranes[12]. However, the intercalation of ions within the graphene oxide membrane is very complex to understand due to the existence of both pristine and oxidized regions, let alone the issue of uncontrolled swelling. The study also investigated the nature of interactions between intercalants and graphene oxide channels[12], especially the cation-π interactions, albeit without much success. Recently, there has been a surge of interest to understand the electric field-controlled ion-transport properties of graphene oxide[18,19]. Our study utilizes electric field to intercalate KCl ions and understand the interaction of KCl ions with graphite surfaces.

## Results and discussion

For the present study, the highly oriented graphitic piece was epoxy glued onto an acrylic sheet that had a pre-fabricated hole of size 2 mm × 2 mm (inset of Fig. 1b). The epoxy ensures that the only path for the ion diffusion is the interlayer space. Detailed information on sample preparations can be found in Supplementary Note 1. Before the intercalation process, we tried to measure the leakage current through this sample. Aqueous NaCl solution of concentrations from $10^{-5}$ M to 1 M was used for the measurement. The leakage current for all the concentrations was

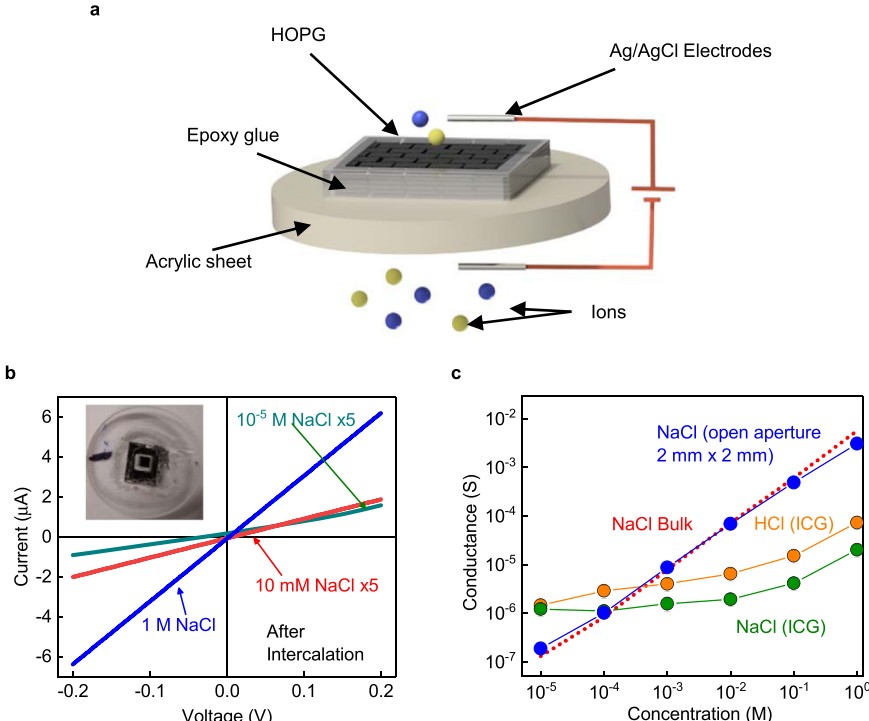

**Fig. 1 Ion-transport through intercalated graphite (ICG). a** Schematic of our ion-transport measurement setup. **b** I–V characteristics of ICG with an equal concentration of NaCl in both reservoirs. The top inset shows an actual ICG device. NaCl concentrations vary from $10^{-5}$ M to 1 M. **c** Conductance of ICG for both HCl and NaCl, along with the data from the open aperture support (blue spheres) of area 2 mm × 2 mm and the effective thickness of 0.25 mm. Error bars provide the standard deviation between repeated measurements. The red dotted curve corresponds to standard data taken from literature[31]. The conductance of ICG at 1 M NaCl concentration is ~3 orders of magnitude smaller than that of bulk.

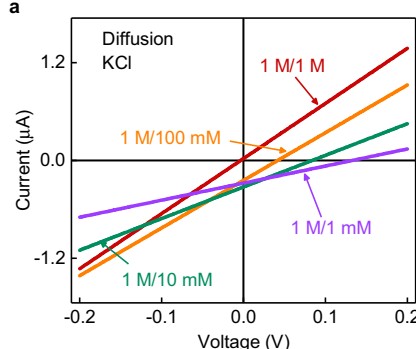
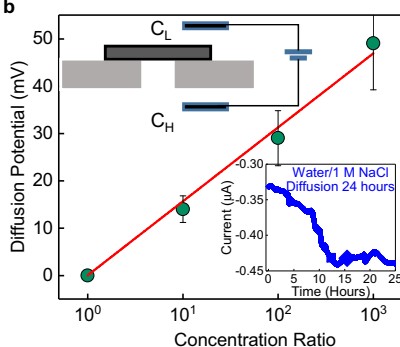

**Fig. 2 Poor ion selectivity of intercalated graphite. a** Current–voltage characteristics for various concentrations of KCl under a concentration gradient. The concentration of one of the reservoirs ($C_H$) was fixed to 1 M and the other ($C_L$) was varied from 1 M to 1 mM. **b** The estimated diffusion potential is plotted as a function of the concentration ratio. The red solid line indicates the best fit to the measured data. The error bar provides an estimate of the spread in the diffusion potential in repeated measurements. The top inset shows the schematic of the drift-diffusion experiments. The bottom inset shows the diffusion current measured over a duration of 24 h, with water on one side of the sample and 1 M NaCl on other side.

less than $10^{-10}$ A at an applied voltage of ±100 mV (conductance ~$10^{-9}$ S). Further details about the measurement can be found in the Methods section and Supplementary Note 2.

The as-prepared samples were then taken for electrochemical intercalation in a setup shown in Fig. 1a & Supplementary Fig. 1. For the intercalation, aqueous solution of 2 M KCl was used. A voltage was applied across the platinum electrodes, which is expected to enhance the ion diffusion through the graphite sample. The diffusion process depends on both the magnitude of voltage and its time duration, and accordingly, we have done several iterations to optimize these values. The applied voltage was varied in the range of 2–10 V and the intercalation time from 2 to 12 h. We in-situ monitored the resulting ionic current through the sample, and the successful intercalation is indicated by a sudden increase in ionic current. We also noticed that at this point, graphite begins to exfoliate, and simultaneously the surface becomes rough. A voltage of 10 V for a duration of 3 h was found to be the optimum for samples of 0.25–1 mm thickness. The ion intercalation is also evident from the samples' recorded X-ray diffraction pattern that shows a new feature at $2\theta = 9.10°$, corresponding to an interlayer distance of 0.97 nm. In addition, the intense peak (0 0 2) has split into two peaks, and most of the higher angle peaks disappeared, suggesting modification in the graphite crystalline structure (see Supplementary Fig. 2). To further confirm this modification, we performed water contact angle measurements. Upon intercalation, the graphitic surface becomes hydrophilic as evident from the reduced contact angle of ~70° from ~104° (see Supplementary Fig. 3). The smaller contact angle after the intercalation process could be partially related to the increased surface roughness. To find the contact angle of inner layers, we removed several layers from the top side of the sample. In this case, the contact angle is ~84°, a value close to the reported result from graphene fluidic channels[20]. This measurement suggests that the intercalated graphite (ICG) channel is hydrophilic at the entrance/exist and nearly hydrophobic in the bulk of the channel.

Immediately after the intercalation, the current–voltage (*I–V*) characteristics were measured using equal molar concentration, *C* of NaCl in both the reservoirs. The intercalation clearly shows increased ionic current through the graphitic sample (Fig. 1b). At ±100 mV and a concentration of $10^{-5}$ M, the measured current is of the order of ~$10^{-7}$ A. This concentration roughly corresponds to that of ions in water as the measured pH of our DI water is ~5.5. The corresponding water conductance is ~$10^{-6}$ S, which is three orders of magnitude larger than those measured for unintercalated graphite. The estimated conductance remains

nearly constant in the concentration range of $10^{-5}–10^{-2}$ M (Fig. 1c). These measurements were repeated using other salt ions of different cationic hydrated diameters and valences. The data for HCl and other salts is shown in Figs. 1c and 4a, respectively. Different salt ions show similar constant conductance in the concentration range of $10^{-5}–10^{-2}$ M with weak dependency on valence and hydrated diameter. Beyond $10^{-2}$ M, the NaCl conductance increases gradually, and at 1 M concentration, the increase is a factor of 10–20 times that of water. Interestingly, at this concentration of 1 M, the conductance of all these salts shows bulk-like behavior as inferred from the ratio of conductances (Fig. 4a). This conductance enhancement through intercalated graphite (ICG) sample is really small when compared to bulk samples where the latter exhibit 5 orders of increase in ionic conductance when the concentration is varied from $10^{-5}$ to 1 M. This strongly indicates high salt rejection efficiency of ICG samples, even at the practically relevant sea salt concentration of 0.6 M. A histogram showing the conductance distribution across several samples is shown in Supplementary Fig. 4a. When the thickness of sample is reduced, an increase in conductance is observed (Supplementary Fig. 4b).

Next, we tried to find out what fraction of ions are transported through the interlayer spaces. We measured the ionic conductance of NaCl through the same acrylic support with a 2 mm × 2 mm hole in it, but this time without the graphite. A comparison of the conductance data with and without graphite shows that at 1 M concentration, the ionic conductance through graphite has been suppressed by 3 orders of magnitude. This suggests that even at the highest concentration of 1 M, the actual ion concentration transported through the sample would be smaller at least by $10^{-3}$ M. We tried to measure the actual concentration of salts that are transported through graphite with the help of diffusion experiments. In this experiment, the two reservoirs, high concentration ($C_H$) and low concentration ($C_L$) were filled with 1 M NaCl and water, respectively. We continuously monitored the diffusion current and a steady state is attained in 24 h. The diffused ions from the $C_L$ side, if any, were collected (inset of Fig. 2b) and examined using inductively coupled plasma (ICP)-mass spectrometer (MS). The osmotic pressure forces an equivalent amount of water from the $C_L$ side to $C_H$ side. Therefore, this measurement provides an upper-bound estimate on the Na concentration, which is ~18 ppm. This is equivalent to $7 \times 10^{-4}$ M of Na, which agrees with the result of the conductance measurement. The volume change in our case is rather small for a duration of 24 h[21], which allowed us to estimate the salt rejection efficiency as $1 - C_L/C_H$, which is more than 99%.

To find out more about the role of interlayer spaces in transport, we performed drift-diffusion measurements. The reservoirs were filled with different concentrations of aqueous KCl solutions. We choose KCl due to similar diffusivities of $K^+$ and $Cl^-$ that ensures zero contribution from liquid junction potential to the measured potential. In these measurements, the concentration of $C_H$ was fixed to 1 M, and the $C_L$ to 1 mM, 10 mM, and 100 mM and 1 M. In the absence of any applied voltage, any difference in the diffusion rates of cations and anions appears as a finite current, consequently, the $I–V$ curves shift along the voltage axis (Fig. 2a). For our configuration (Inset of Fig. 2b), the negative current at zero potential suggests higher diffusion rates for cations compared to anions. The potential corresponding to zero current, $V_0$ is estimated from the $I–V$ curves (Fig. 2a). We estimated the diffusion potential, $V_{diff} = V_0–V_R$, where $V_R$ is the redox potential. $V_{diff}$ is found to exhibit a logarithmic relationship with the concentration ratio (Fig. 2b), which can be described by the Nernst equation[22]. This allowed us to estimate the ion selectivity, $S$ as described below

$$V_{diff} = S\frac{RT}{F}\ln\left(\frac{C_H}{C_L}\right) \qquad (1)$$

Here, $S = t_+ – t_-$, where $t_+$ and $t_-$ are the transport number of cations and anions, respectively. $S = 1$, for the ideal cation-selective channels and $S = 0$, for the non-selective channels. $R$, $T$, and $F$ are the Universal gas constant, the temperature, and the Faraday constant, respectively. The selectivity in our case is found to be 0.26, which suggests weak selectivity between cation and anions. This observation along with the constant conductance at lower salt concentrations, at first instance, might probably indicate a constant surface charge governed regime, however as we discuss later, the constant conductance is most likely arising from steric exclusion.

To understand the nature of the transport of water molecules through interlayer spaces, we performed evaporation experiments (for details, see Supplementary Note 3) using a high-precision gravimetric setup (see Supplementary Fig. 5). The schematic of the setup is shown in Fig. 3a. The weight loss as a result of evaporation through the sample is monitored as a function of time. No significant weight loss was detected through unintercalated graphite samples, for more than 12 h (Fig. 3b). On the other hand, the intercalated graphitic samples showed large weight loss sufficiently larger than those measured from a similar sized open aperture. To quantify the evaporation rate, $Q$, we determined the slope of the weight loss with respect to time. For the intercalated graphite samples that we used in our ion-transport measurements, $Q \approx 1\,\mu g\,s^{-1}$. We observed that $Q$ is highest and more or less constant if the samples are in direct contact with water. We found no effect on $Q$ for different levels of relative humidity. The active area, $A$, that is responsible for $Q$ was estimated from the ionic conductivity data, which comes out to be 3.4 mm$^2$, close to that measured using an optical microscope. Unlike previous studies, our water conductivity experiments allowed us to estimate the effective area very accurately. We utilized this $Q$ to find the flux by considering the active area of the samples, which is ~1 L m$^{-2}$ h$^{-1}$. We additionally performed forward osmosis experiment, using sucrose on one side of the ICG and water on the other side to estimate the water permeation flux. The measured volume change and the estimated water flux comes out to be very similar to that calculated from water evaporation measurements. This is not surprising given that in both experiments, water flows as a liquid. More details on the forward osmosis experiment is provided in Supplementary Note 4.

Several studies on water flow through narrow hydrophobic carbon nanostructures indicate that water remains in liquid state inside these structures[20,23,24]. The classical flow equations failed to explain the high flow rate through our samples (see Supplementary Note 5). So, we checked the possibility of slip-flow with the help of Hagen-Poiseuille equation with slip-boundary conditions. We estimated the pressure difference ($\Delta P$) that is responsible for the high evaporation rates as

$$\Delta P = \frac{12Q\eta L}{\rho h^3 w}\left[1 + \frac{6\delta}{h}\right]^{-1} \qquad (2)$$

where $\eta$ and $\rho$ are the viscosity and the density of water, respectively. $L$, $w$, and $h$ are the length, width, and the interlayer spacing, respectively. Here, $L = 0.25$ mm, $w = 2$ mm. The channel height ($h$) that is available for the passage of water molecules is taken to be 6.3 Å, which is obtained by subtracting the thickness of the carbon sheet (3.4 Å) from the measured interlayer spacing (9.7 Å). $\delta$ is the slip length, which is taken to be 10 nm, as reported for carbon nanotubes[20,23]. The areal density of the channels is estimated from the relation $A/(w \times h)$. The estimated pressure is ~65 bar, a value smaller than previous reports[20,24]. High water flux is also reported in rGO membranes with an interlayer spacing of 0.37 nm[25], where the fast flow is attributed to hydrophobic graphitic regions. We also observe a very similar fast flow through our samples, albeit with high salt rejection. We, however, stress here that high water evaporation rate was not observed in the unintercalated or pristine graphite samples, which clearly indicate the importance of intercalation in our intercalated samples.

To confirm that potassium and chlorine are indeed present in the interlayer spacing, EDAX-SEM analysis of the samples were done, both before and after the intercalation (see Supplementary Note 6). The pristine graphite sample was of high purity and as expected we could detect only carbon before the intercalation

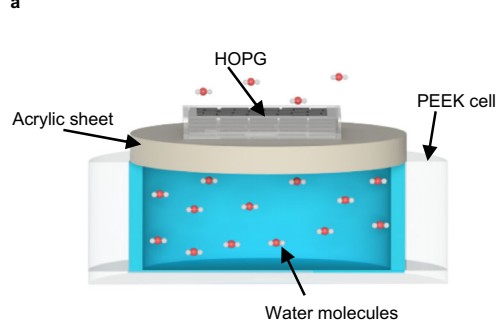

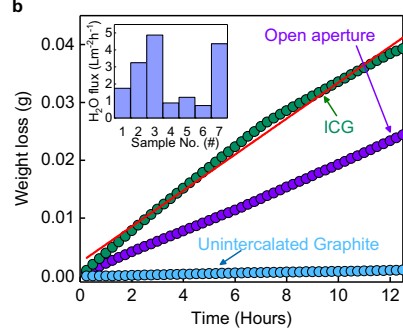

**Fig. 3 Water permeation through intercalated graphite. a** Schematic of our water evaporation setup. **b** Weight-loss through graphite samples is monitored as a function of time, which is a measure of the water evaporation rate. The area of the sample is 3.4 mm$^2$. Over a span of 12 h, the measured weight-loss was little for the unintercalated graphite sample, whereas the intercalated graphite sample shows weight loss more than an open aperture. The solid red line is the best fit to the measured data. Inset: water evaporation flux measured from seven such samples of ICG.

process. Immediately after the intercalation, we performed cross-section analysis of the graphite sample. For this, the sample was washed several times with deionized water and IPA for removing any residual salt that is on the surface. After this, the sample was sliced from the middle and the middle surface was utilized for the cross-section analysis. In the EDAX-SEM, we could detect both potassium and chlorine in the intercalated graphite sample, in addition to the parent carbon (see Supplementary Fig. 6). An elemental mapping indicates that both K and Cl are uniformly distributed over the surface.

To find out if there is any chemical change in the graphite structure due to intercalation, X-ray photoelectron spectroscopy (XPS) analysis of samples was carried out (see Supplementary Note 6). In the graphite samples that were not intercalated, we mostly detected carbon (91.3 atomic %) along with some oxygen (8.7 atomic %) (see Supplementary Fig. 7), though the origin of oxygen is not very clear. After the intercalation of KCl, carbon (91.89 atomic %), oxygen (2.64 atomic %), potassium (4.77 atomic %), and chlorine (0.7 atomic %) could be detected in the graphite sample. XPS was also carried out at different depths (see Supplementary Fig. 8) to understand the distribution of potassium and chlorine across the sample. We found that the amount of salt ions are similar at different depths, indicating its uniform distribution, in agreement with EDAX results. This result further strengthens our observation that graphite intercalation is successful. The presence of potassium is more favorable inside the graphite due to cation-π interactions[12,26], however, we could still detect chlorine which might be an evidence of less-discussed anion-π interactions[26].

A very recent DFT calculation[27], predicts that both ions are favorably incorporated in the interlayer spaces, which agrees with our experimental observation. The samples are highly water stable without any evidence of swelling and the characteristics were unaffected even after storing in water for several months. The results are highly reproducible, though slight variations in conductance between different samples were observed due to variations in the geometrical parameters.

Having discussed the stability of these samples, we will now discuss in detail the transport characteristics at higher salt concentrations. At a concentration of 1 M, we observe low but distinct conductance for various salts (Fig. 4a), although the actual concentration of the ions transported is smaller than $10^{-3}$ M. Such a low concentration of the transported ions suggest that the ions are rejected due to steric exclusion. The conductance data on various salts at 1 M concentration, interestingly allow us to probe

further the nature of the steric exclusion. The small but finite ionic conductance probably already indicates the soft nature of the hydration shells instead of the commonly discussed hard ball model. We observe that the conductance of the salts decreases in the order; KCl > NaCl > CaCl₂ > MgCl₂. The conductance between salts shows a much larger difference than expected for bulk solutions. For example, in our samples, the conductance ratio of KCl and MgCl₂ is ~3.0, whereas in bulk, it is only ~1.4. The data suggests that the conductance of the ions in interlayer spaces is most likely controlled by its (de)hydration energy barrier. To confirm this, we estimated the conductance of salts with respect to HCl conductance and plotted this conductance ratio on a logarithmic scale along with the cation (de)hydration energy of different salts, which is shown in Fig. 4b. Here, the (de)hydration energy is the energy required to completely remove the hydration shells of an ion. We clearly observe a one-to-one correspondence between salt conductance and its (de)hydration energy, and hence further confirm the soft nature of the hydration shells in agreement with previous reports from Jain et al.[4] and Esfandiar et al.[28]. These observations strongly argue that at lower concentrations, the observed constant ionic conductance is most likely a result of steric exclusion and not related to surface charge.

This result also suggests that it is nearly impossible to achieve 100% salt rejection and the soft nature of the ions impose a fundamental limitation. Experimental demonstration of steric exclusion is rarely reported[4,8,28], as it requires fluidic channels of dimensions smaller than ion hydration sizes, where the latter is typically larger than 6.6 Å. The demonstration of steric exclusion in our case is a result of the smaller interlayer spaces with a height of ~6.3 Å. Our study thus provides a foundation for ion separation that relies on hydration energies.

The estimated water flux of ~1 L m⁻² h⁻¹ is comparable to 5–10 L m⁻² h⁻¹ that typically achievable in the forward osmosis set up. It is worth mentioning here that the thickness of our sample is only 0.25 mm, and there is plenty of room at the bottom to improve the water flux. We also observed that several samples showed water flux in the range of ~1–5 L m⁻² h⁻¹ (inset of Fig. 3b), though we confined our discussion to those samples that showed the lowest and exhibited the most reproducible transport properties among them. With high-water flux rate, stability and greater than 99% salt rejection efficiency makes the intercalated graphite an ideal candidate for desalination applications.

To better understand the role of KCl ions in enhancing the ionic conductance through interlayer spaces of graphite, we used

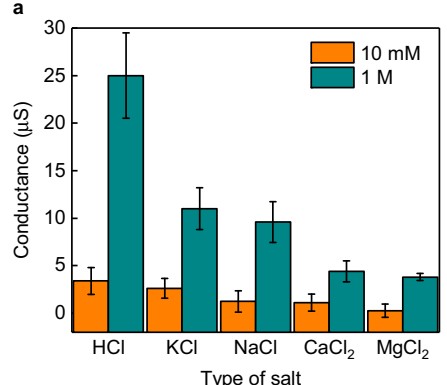
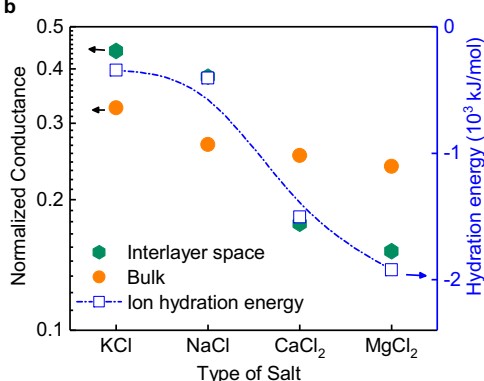

**Fig. 4 Steric exclusion and soft type transport through intercalated graphite. a** Conductance of several salts for concentrations of 10 mM and 1 M (error bar is calculated from the data of 4 different samples). At lower concentrations, all the salts show very similar conductance, however, at higher concentrations (1 M), the conductance decreases with increase in (de)hydration energy of cations. **b** Plot of conductance normalized with respect to HCl for various salt solutions, through ICG and an open aperture. The (de)hydration energy of cations, taken from literature[17] is plotted along the right Y-axis as indicated by blue square symbols. Here, (de)hydration energy is the energy required to completely remove the hydration shells of an ion. The blue dotted line is a guide to the eye. There is one to one correspondence between salt conductance and (de)hydration energy.

deionized (DI) water as the intercalant instead of aqueous KCl. We kept all the intercalation parameters same as that of KCl (see Supplementary Note 7). With water as the intercalant, there was only a little increase in ionic current through the graphite samples (see Supplementary Fig. 9). The small enhancement hints toward the importance of ions in the intercalant solution. The presence of $H_3O^+$ and $OH^-$ ions in DI water seems to help the intercalation process, though with very little efficiency, even at the highest applied voltages of 10 V. Visibly, water intercalated samples had a smooth surface morphology in comparison to KCl intercalated samples, where the latter showed a rough surface after the intercalation process. This experiment also helped us to completely rule out the possibility of any contribution from defects that might be created as a result of high applied voltages. Further, high water flow rates suggest the importance of hydrophobic surfaces, and also the higher water conductance and salt rejection suggest very little possibility of defects dominating the transport.

Having discussed the transport of water molecules and ions through millimeter-sized graphite samples, we now discuss several approaches to its scalability. As the basic building block of these transport channels are graphene layers, thin and large area graphene membranes can be prepared via liquid-phase exfoliation in suitable solvents, followed by vacuum filtration or spin coating. Several studies have already indicated this possibility[29,30]. This approach is very similar to graphene oxide (GO) membrane fabrication. Additionally, graphene membranes can also be prepared by transferring single-layer CVD-grown graphene on top of each other, leading to the possibility of wafer-scale devices.

In summary, an alternate and successful route for the electrochemical intercalation of KCl in graphite is demonstrated that yields high salt rejection efficiencies and enhanced water permeation rates. XPS and EDAX-SEM analysis clearly showed homogeneous distribution of K and Cl inside graphitic interlayer spaces with evidence for dominant cation-π interactions along with non-negligible anion-π interactions. The weak cation vs anion selectivity and low salt conductance is a clear indication that the transport is sterically limited. Bulk-like transport at 1 M concentration suggests a soft ball model of ion transport and the selectivity among salts is determined by their hydration energy. The observed water evaporation rate is successfully explained based on the Hagen–Poiseuille equation for liquid water flow, with slip boundary conditions. High water to ion selectivity and mechanical stability make ICG very useful as a barrier film for filtration processes, and dehumidification applications.

## Methods

**Ion transport measurement**. The small graphitic piece was glued (Stycast 1266 Epoxy, part A and B) onto an acrylic sheet that had a pre-fabricated hole of size 2 mm × 2 mm. A Keithley 2614B source meter and Ag/AgCl electrodes were used for the measurement of current. Reference electrodes (HANA instruments, USA) were also used to verify the measured results.

**Water evaporation measurement**. The measurement setup consists of intercalated graphite samples epoxy sealed on top of a miniature container filled with water. The whole setup was placed in a METTLER TOLEDO high precision balance and the weight loss was monitored as a function of time. The precision of the setup is 10 μg. The relative humidity (*RH*) inside the balance was controlled to <20% in all the measurements.

## Data availability

All data that is required to understand the conclusions in the paper is presented in the main and supplementary information. Additional data related to this paper are available from the corresponding author upon request. Source data are provided with this paper.

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

## Acknowledgements

We acknowledge the financial support received from MHRD STARS with Project no. MoE-STARS/STARS-1/405. We acknowledge the generous access to the central instrumentation facility, IIT Gandhinagar, in executing this work.

## Author contributions

G.K. conceived the idea and supervised the project. L.S. executed the sample preparation, characterization, ion transport, and water evaporation measurements. S.S.N. carried out

the initial optimization of intercalation, ion transport, and XPS measurements. A.R. and S.K. helped in the ion transport and water evaporation measurements. L.S. and G.K. wrote the manuscript. All the authors discussed the results and commented on the manuscript.

## Competing interests

The authors declare no competing interests.
