## [Peer Review File · Nature Communications]

Selective transport of water molecules through interlayer spaces in graphiteREVIEWER COMMENTS

Reviewer #1 (Remarks to the Author):

In this work, the authors described their work on expanded the graphite interlayer spacings using K^+ by electrochemically methods in aqueous solutions, and the water conductivity shows several orders of enhancement when compared to unintercalated graphite. This work is interesting. However, more additional evidences should be provided to strongly support the statements.

The comments are as follows.

1. The authors should cite and analyzed some related work in recent three years, especially for electrochemically methods on the water or ions diffusivity (such as Science, 2021, 372, 501; Nature, 2018, 559.235). I find the refs in this version are all before 2018.
2. Please compare the electrochemically methods to intercalated ions with the methods just using cation themselves (Nature, 2017, 550, 380) and others.
3. To further confirm the process of water permeance of the membranes with or without intercalated, the cycles of drying (evaporating) and wetting performance ARE essential, and the underlying mechanism should be elucidated (including XRD)
4. The calculation of water permeance according evaporation is not standard method in the membrane process. The author should add the common methods to compare the water permeance (ACS Nano, 2021, 15, 6, 9201).
5. Long-term stability is one of the important parameters for membrane performance.
6. Some characterization such as XPS spectra with the corresponding depth profiles should be used to explain the change of the membrane.
7. There are some typos in the manuscript.
8. Explain the "hydration energy" in Fig 4 b.

Overall, I cannot recommend this manuscript to be accepted at the current stage.

Reviewer #2 (Remarks to the Author):

This manuscript (NCOMMS-21-33787) reports on the selective transport of water molecules through KCl intercalated graphite prepared by electrochemical intercalation. The most remarkable claim in the paper is that the KCl intercalated graphite has water permeation rate with weak ion permeation, due to size exclusion (interlayer spacing of $\sim 3.4 \text{ \AA}$). It's an original and interesting work both in experimental control of the interlayer spacing and theoretical analysis in water permeation rate and desalination. However, the main challenge of graphene-based filtration membrane for desalination, is a low water flux when the ion rejection rate is large enough, which limits the practical applications of graphene-based filtration membranes for ions sieving [Journal of Materials Chemistry A. 2021, 9, 10672-10677]. The flux requires enhanced through channel optimization or thickness control, as the prepared graphite has no advantage in water flux of $1\sim 5 \text{ L m}^{-2} \text{ h}^{-1}$ compared to early reports. In addition, the water selective transport should be further confirmed by standard ion permeation tests [Science, 2014, 343(6172), 752–754] or filtration experiments [Nature Communications, 2019, 10, 1253]. There are some of issues needed to be addressed for the possible publication in Nature Communications as follows:

1. In the process of electrochemical intercalation, the successful intercalation or formation of water channels was judged by a sudden increase in ionic current. I'm confused that graphite is a conductor with a conductance of $\sim 200 \text{ S}$, while, the conductance of salt bulk solution is $< 0.01 \text{ S}$.

Why does graphite behave like an insulator in this process? What is the conductivity of graphite before and after KCl intercalation?

2. In Fig. 1c, "Beyond 10⁻² M, the NaCl conductance increases gradually and at 1 M concentration, the increase is a factor of 10-20 times that of water", the high ion concentration in two reservoirs, forms ion osmotic pressure on the water channel, which may further causing the channel to expand similar to the electrochemical intercalation. The stability of the interlayer spacing needs to be analysed in long-term operation and high ion concentration, which is crucial for ion rejection.

3. In the KCl diffusion measurements in Fig. 2a, the I-V curve of 1M/1M pass through the zero, while the other curves with different ratio of concentration show negative currents in several μA . It should be ion diffusion due to ion concentration differences. The water selective transport should be further confirmed by standard ion permeation tests [Science, 2014, 343(6172), 752–754] or filtration experiments [Nature Communications, 2019, 10, 1253].

4. In the theoretical analysis in this paper, it is inaccurate to use 3.4 Å as the space of the actual water channel, which is impermeable due to the thickness of the graphite sheet itself.

5. "L.h⁻¹ .m⁻²" should be changed to "L m⁻² h⁻¹".

Reviewer #3 (Remarks to the Author):

In this work, KCl ions were incorporated into graphite interlayer spaces via electrochemical method to achieve selective transport of water molecules. The metal ion intercalation of graphite through electrochemical method is not new (for instance, Li⁺ intercalation for battery). The hydration energy dependent ion transport has also been reported in literatures. Therefore, the novelty of this manuscript is not adequate. In addition, there also exist some critical issues in this manuscript in experimental design and data analysis.

1. The characterization for membrane structure is unclear.

(1) As observed from the insets of Figure S2, the graphite membranes show obvious expansion after intercalation. Why the XRD patterns don't show shift of peaks?

(2) EDAX-SEM is a rather rough characterization for elemental detection. More precise technique such as XPS is suggested.

(3) Is there any change in the chemical structure of graphite after intercalation such as oxidization?

(4) How about the distribution of metal ions in membrane?

2. Aiming at practical application, the solution-processable method is more favorable for membrane fabrication, such as assembly of 2D materials or solution casting. However, the method reported here is not attractive for application due to the small size, high cost and large thickness.

3. The effective membrane area depends on the epoxy glue instead of the 2mm hole in acrylic sheet. Is it reasonable to compare the current without considering the difference in membrane area?

4. The interactions between KCl and graphite are weak π -cation interactions. How about the stability of as-prepared membranes?

5. Please provide the detailed information of tested membranes, such as size, and thickness. Taking a photo of the tested membrane is preferred.

In summary, the manuscript in the current status doesn't meet the requirements of Nature Communications.

Point by Point response to the Reviewers comments

Reviewer #1

In this work, the authors described their work on expanded the graphite interlayer spacings using K+ by electrochemically methods in aqueous solutions, and the water conductivity shows several orders of enhancement when compared to unintercalated graphite. This work is interesting.

We thank the Reviewer for careful reading of our manuscript and helpful comments.

However, more additional evidences should be provided to strongly support the statements. The comments are as follows.

1. The authors should cite and analyzed some related work in recent three years, especially for electrochemically methods on the water or ions diffusivity (such as Science, 2021, 372, 501; Nature, 2018, 559.235). I find the refs in this version are all before 2018.

We have now added these references in the revised manuscript and added few statements regarding its findings (Lines 45-47).

2. Please compare the electrochemically methods to intercalated ions with the methods just using cation themselves (Nature, 2017, 550, 380) and others.

In the case of graphene oxide controlled by cations (Nature 550, 380 (2017)), the cations preferentially interact with the negatively charged functional groups, though cation- π interactions are also mentioned as possibilities, and the experimental evidence for it is very meek in the paper. XPS was also reported in the paper (Nature 550, 380 (2017)), however, there exist little evidence for K in their samples with almost no evidence for cation- π interactions. In the case of graphene oxide, since the density of functional groups is quite large, the electrostatic interaction of cations with these groups is going to dominate and it is extremely difficult to extract any information on cation- π interactions from these samples.

However, in our study, the absence of functional groups in graphite enabled us to better understand the cation- π interactions and this is very clear from our XPS study (Supplementary figs. 7-8) also. We see clear evidence for the successful intercalation of K and Cl inside these interlayer spaces with characteristic features related to K and Cl, though the atomic percentage of Cl is much lower than K. In addition, we also notice that after the intercalation, the peak corresponding to π - π^* interaction is suppressed strongly suggesting changes in the interlayer interactions.

In summary, our study has shed light on new evidences for cation- π interactions and the intercalation of K and Cl with very good control on the permeation characteristics, with potential applications.

3. To further confirm the process of water permeance of the membranes with or without intercalated, the cycles of drying (evaporating) and wetting performance ARE essential, and the underlying mechanism should be elucidated (including XRD)

We thank the Reviewer for pointing this out. We observe that if we let the membranes dry, the channels get blocked due to adsorption of molecules at the entrance and exit of the channels. To remove these adsorbed molecules, we had to apply a voltage across the membrane, which removes these adsorbents and after this process, the channels are ready for the transport of water molecules. This is usually the case with small sized channels as the hydrophilic entrance/exist tend to adsorb molecules. To avoid the adsorption, we never let dry the sample and always try to store the samples in water. This is also mentioned in a related paper by Keerthi *et al.* Nature 558, 420 (2018), where sub-nm nanochannels are blocked due to molecular adsorption. Also, our estimations clearly indicate that water is transported in the liquid form through the interlayer space and not in the vapor form. If we consider the transport to be of vapor form, then the estimated water flux rate is seven orders smaller than measured (Supplementary section 6). This rule out any possibility of gaseous transport. In the XRD data as well, we did not find any difference in the interlayer distance, in both dry and wet state.

4. The calculation of water permeance according evaporation is not standard method in the membrane process. The author should add the common methods to compare the water permeance (ACS Nano, 2021, 15, 6, 9201).

We agree with the Reviewer and have now performed permeation measurements with water on one side of the intercalated graphite sample and 2 M sucrose on the other side. The measured water permeance rate is comparable to that estimated from the water evaporation measurement. We also attached some camera images to indicate the increase in the level of water on the sucrose side (A paragraph is included in Supplementary section 7 describing this measurement and the results). We also cited this reference.

Fig. R1: Water permeation experiment using water on one side of the ICG and sucrose on the other side. (a) At the beginning of the permeation test, the levels are equal. (b) After 24 hours of permeation, an increase in level is seen on the sucrose side, $\sim 100 \mu\text{L}$, which is as expected. The solutions were taken out and carefully measured to find out the volume change.

5. Long-term stability is one of the important parameters for membrane performance.

We performed measurements of the same sample after a gap of 5 months and found that there is no degradation in the permeation properties. For example, the NaCl ionic conductance through ICG is again measured as a function of concentration (Fig. R2). The conductance behavior is found to be very similar, albeit with a very little decrease in NaCl conductance at all concentrations. The salt rejection efficiency is still $> 99\%$, indicating the stability of these devices. In addition, we have performed water filtration

experiments with devices that were made 6 months back and found very little degradation in the estimated water flux. It is important that these devices should be stored in water for longer stability.

Fig. R2 Data repetition of intercalated sample after a gap of 5 months, which was stored in water.

6. Some characterization such as XPS spectra with the corresponding depth profiles should be used to explain the change of the membrane.

We have now performed detailed XPS characterization of the samples (before and after the intercalation process) along with the depth analysis (Figs. R3 & R4). We found that there is clear evidence for the presence of K along with a small percentage of Cl as well. We believe that the presence of Cl inside the interlayer spaces is a result of charge neutrality requirements. Applied voltage did not induce any oxidation as the oxygen percentage after the intercalation is smaller than before the intercalation. We have added these findings in the revised manuscript (Lines 205-215) and supplementary section 8.

Fig. R3. X-ray photoelectron spectroscopy (XPS) results. (a) XPS survey of graphite sample before and after the intercalation of KCl. The presence of chlorine and potassium is evident in the intercalated samples, while this was absent in the unintercalated samples. The carbon peaks look similar both before (b) and after (c) the intercalation process, except for the absence of π - π^* peak after the intercalation process. The presence of oxygen (d) was detected before and after the intercalation process, though the percentage is smaller after the intercalation. Fully resolved chlorine (e), and potassium (f) peaks in the intercalated sample allowed us to estimate the atomic percentage of potassium and chlorine as 4.77% and 0.7%, respectively.

Fig. R4 – XPS depth analysis. XPS data was collected at different depths of the intercalated graphite samples. XPS survey (a) and depth profiling of several elements found in ICG (b-e). (f) The atomic percentage of various elements present in ICG.

7. There are some typos in the manuscript.

We have now corrected the typos in the revised manuscript, which includes the unit of water permeance.

8. Explain the “hydration energy” in Fig 4 b.

We have now added the meaning of hydration energy in Fig. 4b and lines 234-235 of the revised manuscript.

Reviewer #2 (Remarks to the Author):

This manuscript (NCOMMS-21-33787) reports on the selective transport of water molecules through KCl intercalated graphite prepared by electrochemical intercalation. The most remarkable claim in the paper is that the KCl intercalated graphite has water permeation rate with weak ion permeation, due to size exclusion (interlayer spacing of $\sim 3.4 \text{ \AA}$). It's an original and interesting work both in experimental control of the interlayer spacing and theoretical analysis in water permeation rate and desalination.

We thank the Reviewer for careful reading of our manuscript and providing very constructive comments that helped us to significantly improve the present manuscript.

However, the main challenge of graphene-based filtration membrane for desalination, is a low water flux when the ion rejection rate is large enough, which limits the practical applications of graphene-based filtration membranes for ions sieving [Journal of Materials Chemistry A. 2021, 9, 10672-10677]. The flux requires enhanced through channel optimization or thickness control, as the prepared graphite has no advantage in water flux of $1\sim 5 \text{ L m}^{-2} \text{ h}^{-1}$ compared to early reports. In addition, the water selective transport should be further confirmed by standard ion permeation tests [Science, 2014, 343(6172), 752–754] or filtration experiments [Nature Communications, 2019, 10, 1253].

Our aim is clearly not to beat the records, but to probe the molecule/ion transport through a simple system where the complex functional groups are absent, unlike graphene oxide. There were myriads of papers in the area of graphene oxide and the science is too complex to comprehend due to the presence of several functional groups and swelling in water. Our study considers the simplistic aspect of the graphitic interlayers, with a careful modification of the interlayer space. Hydrophobic channels are predicted to follow Hagen-Poiseuille equation with slip enhanced flow. By taking a simple system of graphite, we could clearly identify the slip enhanced flow, pointing that in graphene oxide, most of the enhancement is related to the graphitic regions and not oxidized regions.

The optimization of thickness is just an engineering issue and not a scientific one. In terms of quality of the device and the understanding of science, graphite is much better than the mechanically assembled graphene oxide layers, where the latter is prone to defects and pinholes. Nevertheless, we have now performed permeation experiments with samples of smaller thickness ($\sim 30 \mu\text{m}$) as well and found that the water conductance proportionally showed increase (Supplementary fig.4). We also observed that the water evaporation rates for thinner samples also showed a proportional increase with a water flux rate of $\sim 20 \text{ L.m}^{-2}.\text{h}^{-1}$ (Fig. R5).

We have already reported the ion permeation experiment in the main text (Lines 107-122). Based on the Reviewer's suggestions, we have now carried out water filtration experiments also and the estimated water flow is very similar to that estimated from the water evaporation experiments (Also see response to Reviewer #1, Question-4). This is not surprising given that in both experiments, water flows as liquid (Supplementary section 6-7). With this result, we have added few statements in the revised main manuscript (Lines 166-170).

Fig. R5: Water evaporation at different thicknesses of the sample. The data shows increased water evaporation rates through thin samples. The estimated water flux rate for the 30 μm thick sample is $\sim 20 \text{ L}\cdot\text{m}^{-2}\cdot\text{h}^{-1}$.

There are some of issues needed to be addressed for the possible publication in Nature Communications as follows:

1. In the process of electrochemical intercalation, the successful intercalation or formation of water channels was judged by a sudden increase in ionic current. I'm confused that graphite is a conductor with a conductance of $\sim 200 \text{ S}$, while, the conductance of salt bulk solution is $< 0.01 \text{ S}$. Why does graphite behave like an insulator in this process? What is the conductivity of graphite before and after KCl intercalation?

Expansion and exfoliation of graphite layers in aqueous solutions require high applied voltages. A literature survey on the electrochemical exfoliation of graphite in the presence of aqueous H_2SO_4 and other sulphate electrolytes (this method is used for graphene oxide synthesis) indicated that the exfoliation typically requires DC voltages of 10 V (Parvez K. *et al.* J. Am. Chem. Soc. 136, 6083 (2014), Wu. *et al.*, Small 10, 1421 (2014), Achee T.C. *et al.*, Sci Rep. 8, 14525 (2018), Huang *et al.*, Nanotechnology 26, 105602 (2015) and many others). The requirement of high voltage is a result of the change in work function of hydrophobic graphite when it is in contact with aqueous solutions. Before the intercalation, there is no space for any molecule or ion to pass through so the measured conductance is very less, and it behaves like a capacitor storing charges on its edges as we clearly see hysteresis in the current-voltage measurements, which makes it even difficult to accurately measure the conductance before the intercalation. The applied voltage probably creates radicals in water, which reduces hydrophobicity and help the intercalation process. The expanded interlayers allow the permeation of water molecules and ions with a concomitant increase in conductance. In another paper of our own, we have measured the conductance of graphene channels (Gopinadhan *et al.* Science 2019, 363, 145) fabricated using exfoliation and lithographic techniques. We observed that if there is no graphene channel, and only graphite pieces, the currents are in the range of pS, indicating that graphite does not alone transport any ions and contribute to the overall conductance.

The NaCl conductivity of graphite after the intercalation is $\sim 3.10^{-4} \text{ S/m}$, which is roughly 3.3 orders of magnitude smaller than the bulk ionic conductivity of NaCl, i.e. 10.28 S/m. It is not possible to calculate the ionic conductivity before the intercalation as the ionic current is much smaller due to the absence of interlayer transport.

We have used contact angle measurements to understand the nature of the surface upon intercalation. Upon intercalation, the graphitic surface becomes hydrophilic as evident from the reduced contact angle of $\sim 70^\circ$ from $\sim 104^\circ$ (supplementary fig. 3). The smaller contact angle after the intercalation process could be partially related to the increased surface roughness. To find the contact angle of inner layers, we removed several layers from the top side of the sample. In this case, the contact angle is $\sim 84^\circ$, a value close to the reported result from graphene fluidic channels (Radha Boya, Nature **538**, 222–225 (2016)). This measurement suggests that the ICG channel is hydrophilic at the entrance/exist and nearly hydrophobic at the bulk of the channel. These statements are now added to the revised main manuscript (Lines 70-77) and also added a figure to the supplementary information file (Supplementary fig. 3).

Fig. R6: Contact angle measurements. (a) The graphite surface is hydrophobic before the intercalation. (b) After the intercalation, the graphite surface becomes hydrophilic. (c) After removing a few top layers of intercalated graphite, the surface is less hydrophobic than the pristine sample.

2. In Fig. 1c, “Beyond 10-2 M, the NaCl conductance increases gradually and at 1 M concentration, the increase is a factor of 10-20 times that of water”, the high ion concentration in two reservoirs, forms ion osmotic pressure on the water channel, which may further causing the channel to expand similar to the electrochemical intercalation. The stability of the interlayer spacing needs to be analysed in long-term operation and high ion concentration, which is crucial for ion rejection.

The ionic conductance is measured with equal salt concentration on both sides of the device and therefore, there is no net osmotic pressure across the device, which rules out any possibility of osmotic pressure expanding the interlayer spaces temporarily. We have now included the result for a representative device that is measured after 5 months though several devices were tested, with no sign of degradation in the performance (see response to Reviewer #1, Fig.R2). It should be noted our ion transport measurements were done upto a salt concentration of 1 M, which is larger than the typical sea salt concentration of 0.6 M.

3. In the KCl diffusion measurements in Fig. 2a, the I-V curve of 1M/1M pass through the zero, while the other curves with different ratio of concentration show negative currents in several μA . It should be ion diffusion due to ion concentration differences. The water selective transport should be further confirmed by standard ion permeation tests [Science, 2014, 343(6172), 752–754] or filtration experiments [Nature Communications, 2019, 10, 1253].

Though we have discussed the origin of very small diffusion currents in the main manuscript in length (Page 4, starting from line 123), let us reiterate here. As the Reviewer rightly mentioned that there appears a very small fraction of ions diffusing through the graphitic interlayer spaces as indicated by the negative diffusion currents. Additionally, we have quantified the diffused concentration of ions at the permeate side with the help of inductively coupled plasma (ICP)-mass spectrometer (MS) technique. However, the measured concentration is really small, and the rejection ratio is still > 99%. In the main text, we explained the origin of this very small diffusion currents to arise from the soft nature of the ion

hydration shells. We have also performed filtration experiments using sucrose and water on either side of the membranes. The estimated water flux rates are comparable to those estimated from water evaporation experiments. In hydrophobic, highly confined channels, the water flows as a liquid than vapors in the evaporation experiment. The agreement between these two measurements further confirms that the evaporation rate can also be used to measure the water permeance in the hydrophobic channels.

4. In the theoretical analysis in this paper, it is inaccurate to use 3.4 Å as the space of the actual water channel, which is impermeable due to the thickness of the graphite sheet itself.

Based on the latest X-ray diffraction data, for the calculations, we have now used a new interlayer distance by subtracting the thickness of the graphene sheet. The X-ray diffraction data provides an interlayer space of 9.7 Å as estimated from the lower angle peak of $2\theta = 9.10^\circ$. By considering the thickness of the graphene sheet, which is ~ 3.4 Å, the practical interlayer space available for the transport is 6.3 Å. This value is smaller than the typical hydration sizes of salt ions used in this study.

5. "L.h-1 .m-2" should be changed to "L m-2 h-1".

We apologize for this typo. We have corrected this in our revised manuscript, both in main and supplementary information.

Reviewer #3 (Remarks to the Author):

In this work, KCl ions were incorporated into graphite interlayer spaces via electrochemical method to achieve selective transport of water molecules. The metal ion intercalation of graphite through electrochemical method is not new (for instance, Li+ intercalation for battery). The hydration energy dependent ion transport has also been reported in literatures. Therefore, the novelty of this manuscript is not adequate. In addition, there also exist some critical issues in this manuscript in experimental design and data analysis.

We thank the Reviewer for the critical and very useful comments. We politely disagree with the Reviewer's comment as our ion intercalation method uses aqueous electrolytes against the metal intercalation method that the Reviewer is pointing out. The latter belongs to non-aqueous electrolytes, which is the reason behind the success of Li-ion batteries. Aqueous-based batteries are increasingly sought after, especially for military applications. Researchers recently (2014 onwards) started looking at aqueous-based Li-ion batteries, especially the team of Chunsheung Wang from UMD and Kang Xu from ARL. Our experiments shed light on the possibility of intercalation and the mechanism of expansion in graphite electrodes. The method can be easily extended to electrically insulating 2D materials also. More research is required in aqueous-based electrolytes, and our research is only going to help this effort.

The hydration energy becomes important only in channels with comparable ion hydration sizes, and its observation requires fluidic structures in the sub-nm ranges. Recently, there has been some success in

this direction; however, there is no similar study of water permeation and soft-ball model of ion transport. We would also like to stress that hydration energy is not the only important finding of this study.

1. The characterization for membrane structure is unclear. As observed from the insets of Figure S2, the graphite membranes show obvious expansion after intercalation. Why the XRD patterns don't show shift of peaks?

We thank the Reviewer for pointing this out. Though there is a visible expansion of the graphite after the intercalation process, XRD was somehow unable to detect any peak shift. However, we have repeated those measurements with different XRD parameters. After the intercalation, the XRD data (Supplementary Fig. 2) reveals a new peak at $2\theta = 9.1^\circ$, corresponding to an interlayer distance of 9.7 Å. After subtracting a carbon height of 3.4 Å, the available interlayer distance is 6.3 Å, which is sufficient for the transport of water molecules while rejecting the salt ions. However, we observe from different methods of electrochemical intercalation of graphite, XRD data was unable to provide complete information about the interlayer changes. At the same time, XPS was much more sensitive and provided vital information about the intercalation process.

2. EDAX-SEM is a rather rough characterization for elemental detection. More precise technique such as XPS is suggested.

We have now carried out XPS studies, and the results confirm the conclusion derived from EDAX that the intercalants are uniformly distributed across the sample. More details is given in response to Reviewer #1, Q.6.

3. Is there any change in the chemical structure of graphite after intercalation such as oxidization?

We did not detect any chemical changes from XPS studies. See response to response to Reviewer #1, Q.6.

4. How about the distribution of metal ions in membrane?

Even distribution is evident from the EDAX (Supplementary fig. 6) and XPS results (Supplementary Fig. 7-8).

5. Aiming at practical application, the solution-process able method is more favorable for membrane fabrication, such as assembly of 2D materials or solution casting. However, the method reported here is not attractive for application due to the small size, high cost and large thickness.

The intercalated graphite is robust and not prone to breakage, or cracks while handling, while the membrane prepared from solutions is subjected to pin-holes and defects. There is almost consensus that the transport is mostly pin-hole or defect-dominated in those solution-prepared membranes. Thinning the graphite is merely an engineering issue and not a fundamental limit.

6. The effective membrane area depends on the epoxy glue instead of the 2mm hole in acrylic sheet. Is it reasonable to compare the current without considering the difference in membrane area?

We measured the sample area with the help of an optical microscope, and it is $\sim 3.2 \text{ mm}^2$, very close to the 4 mm^2 data we mentioned for the bulk data. The estimated ionic conductance is hardly affected by this slight difference.

7. The interactions between KCl and graphite are weak π -cation interactions. How about the stability of as-prepared membranes?

Yes, we observe that there is cation- π interactions (see response to Reviewer #1). The as-prepared samples show good stability even after five months of storage in water (Fig. R2).

8. Please provide detailed information of tested membranes, such as size, and thickness. Taking a photo of the tested membrane is preferred.

We have tested samples having different areas ranging from $1\text{-}15 \text{ mm}^2$ and thickness ranging from $30 \mu\text{m}$ to 1 mm .

Sample No.	Water Conductance (S)	Sample length (m)	Active area in (m^2)
1	6.74E-08	1.00E-03	1.35E-06
2	8.94E-08	3.00E-05	5.36E-08
3	2.67E-07	2.50E-04	1.34E-06
4	4.40E-07	2.50E-04	2.20E-06
5	4.90E-07	2.50E-04	2.45E-06
6	5.72E-07	2.50E-04	2.86E-06
7	6.03E-07	2.50E-04	3.02E-06
8	7.03E-07	2.50E-04	3.52E-06
9	8.01E-07	2.50E-04	4.01E-06
10	8.09E-07	2.50E-04	4.05E-06
11	9.59E-07	2.50E-04	4.80E-06
12	1.31E-06	5.00E-04	1.31E-05
13	1.77E-06	2.50E-04	8.85E-06
14	2.34E-06	2.50E-04	1.17E-05

Fig. R7: Some images of tested samples and the water conductance measured across them.

REVIEWERS' COMMENTS

Reviewer #1 (Remarks to the Author):

The authors well respond my questions and now I suggest its publication.

Reviewer #2 (Remarks to the Author):

The authors addressed all of my comments. I have no further comments on this manuscript.

Reviewer #3 (Remarks to the Author):

The authors have added significant amounts of text in response to the comments made by the three reviewers. I appreciate the effort. There are still some issues need to be addressed before considering for publication.

1. For the comment 4 of reviewer #1, the authors performed water permeance measurement via taking out the solutions. This process may lead to great inaccuracy since the total volume change is only 100 μL . The direct measurement of volume change without the taking out process is a standard method with higher precision (Nano Lett. 2015, 15, 3254–3260).

2. The authors mentioned that the metal intercalation method is based on aqueous electrolyte, while the literatures belong to non-aqueous electrolytes. Theoretically, there is no big difference between aqueous and non-aqueous environment for ion intercalation. The energy barrier of ion intercalation in water could be higher, but that can be overcome by higher voltage.

3. As for the practical application of this membrane, the authors claimed that thinning the graphite is merely an engineering issue. Is it possible to prepare graphite membrane with nm thickness? From the perspective of practical membrane fabrication, it would be a tough challenge to prepare nm-thick graphite film with large area. In addition, how about the small size and high cost issues? What is the scale-up potential of this membrane? Currently, 2D GO membranes with dozens of centimeters in size have been prepared with various methods and show excellent performance.

Point by Point Response to the Reviewers comments

Reviewer #1

The authors well respond my questions and now I suggest its publication.

We thank the reviewer for the critical and valuable comments, which helped us significantly improve the manuscript.

Reviewer #2:

The authors addressed all of my comments. I have no further comments on this manuscript.

We are very thankful to the reviewer for the positive and constructive suggestions.

Reviewer #3:

The authors have added significant amounts of text in response to the comments made by the three reviewers. I appreciate the effort. There are still some issues need to be addressed before considering for publication.

We are very thankful to the reviewer for the positive, constructive, and critical suggestions. We provide below our detailed responses.

1. For the comment 4 of reviewer #1, the authors performed water permeance measurement via taking out the solutions. This process may lead to great inaccuracy since the total volume change is only 100 μ L. The direct measurement of volume change without the taking out process is a standard method with higher precision (Nano Lett. 2015, 15, 3254–3260).

We agree with the reviewer that there could be inaccuracies in volume measurement in the solution removal method. We have utilized high-precision micro-pipettes to minimize possible errors and measure tiny volumes with high accuracy. We also observed changes in water levels through camera images; however, the accurate estimation of small changes in volume was difficult as the magnitude of the change was around 100 μ L. To address this issue and increase the water flux, we plan to fabricate large-area graphene membranes via the liquid-phase exfoliation of graphite with suitable solvents, similar to graphene oxide membrane fabrication (see the response to Q.3).

Few statements related to this has been added to the revised supplementary file on page 8.

2. The authors mentioned that the metal intercalation method is based on aqueous electrolyte, while the literatures belong to non-aqueous electrolytes. Theoretically, there is no big difference between aqueous and non-aqueous environment for ion intercalation. The energy barrier of ion intercalation in water could be higher, but that can be overcome by higher voltage.

Our experiment agrees with the theoretical expectation that aqueous solutions should be harder to intercalate, and a higher voltage is required for successful intercalation. However, high applied voltages

to material systems might also have associated problems, like oxidation, which was also another exciting direction we have pursued. Probing the cation/anion- π interactions in a clean system like graphite is also an important research problem that we have investigated.

A statement related to cation/anion- π interactions has been added to the abstract and the conclusions as this is an important finding.

3. As for the practical application of this membrane, the authors claimed that thinning the graphite is merely an engineering issue. Is it possible to prepare graphite membrane with nm thickness? From the perspective of practical membrane fabrication, it would be a tough challenge to prepare nm-thick graphite film with large area. In addition, how about the small size and high cost issues? What is the scale-up potential of this membrane? Currently, 2D GO membranes with dozens of centimeters in size have been prepared with various methods and show excellent performance.

Thin, thick, and large area graphene membranes can be prepared via liquid-phase exfoliation with suitable solvents, followed by vacuum filtration or spin coating. Several studies have already indicated this possibility (Y. Hernandez *et al.* Nat. Nanotech. 3, 593 (2008), V. Nicolosi *et al.*, *Science* 340, 1226419 (2013)). This approach is very similar to graphene oxide (GO) membrane fabrication. The cost could be slightly smaller than GO's as this process does not require any strong oxidizing agents and only needs cheaper salts for the intercalation. With thin membranes, the duration of KCl intercalation is also short, in the range of a few seconds. Additionally, graphene membranes can be prepared by transferring single-layer CVD-grown graphene on top of each other, leading to the possibility of wafer-scale devices.

Few statements related to the scalability of graphene membranes has been added to the revised main manuscript on page 6, just before the conclusions paragraph.